# Functional Differences between Chewing Sides of Implant-Supported Denture Wearers

**Masaoki Yokoyama [1], Hiroshi Shiga [1,*], Shin Ogura [2], Mako Sano [1], Marie Komino [1], Hitoshi Takamori [2], Hanako Uesugi [1], Keiji Haga [1] and Yoshikazu Murakami [1]**

[1] Department of Partial and Complete Denture, School of Life Dentistry at Tokyo, The Nippon Dental University, 1-9-20 Fujimi, Chiyoda-ku, Tokyo 102-0071, Japan

[2] Division of Oral Implant, Hospital at Tokyo, The Nippon Dental University, 2-3-16 Fujimi, Chiyoda-ku, Tokyo 102-8158, Japan

[*] Correspondence: h-shiga@ndu.ac.jp; Tel.: +81-33-261-5729

**Abstract:** Humans are said to have habitual and non-habitual chewing sides; however, the functional differences between the chewing sides of implant-supported denture wearers have not been sufficiently clarified. This study aimed to clarify the presence or absence of functional differences between the chewing sides in implant-supported denture wearers. Forty-five patients with bilateral posterior implants were included in this study. The participants were asked to chew a gummy jelly on one side, and the masticatory movement was recorded using a Motion Visi-trainer (MVT V1). For 10 cycles from the fifth cycle after the start of mastication, the pattern of the movement path, the opening distance, the masticatory width, and the cycle time were calculated as parameters of masticatory movement. The amount of glucose eluted during the chewing of gummy jelly was measured and used as a parameter of masticatory performance. Each parameter representing masticatory movement and masticatory performance was compared between the right and left chewing sides and between the habitual and non-habitual chewing sides using a chi-squared test or a paired t-test. There was no difference in the frequency of masticatory path patterns between the right and left chewing sides. Most participants had a normal pattern on the habitual chewing side; however, abnormal patterns were also observed on the non-habitual chewing side. When comparing right and left chewing, no significant difference was observed between chewing sides in terms of opening distance, masticatory width, cycle time, or amount of glucose eluted ($p > 0.05$). When comparing the habitual and non-habitual chewing sides, masticatory movement on the habitual chewing side showed a larger opening distance ($p < 0.001$) and masticatory width ($p = 0.008$), shorter cycle time ($p = 0.004$), and higher masticatory performance ($p < 0.001$). It was suggested that there is a functional difference between the habitual and non-habitual chewing sides in the masticatory movement and masticatory performance of implant-supported denture wearers.

**Keywords:** dentistry; denture wearer; implant-supported denture; masticatory movement; cycle time; masticatory performance

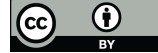

## 1. Introduction

The timing of functional assessment after wearing removable or implant dentures has not been extensively investigated [1]. Sufficient adjustment is necessary to acquire good masticatory function after wearing new dentures and functional evaluation should be performed after acclimatization to dentures [2,3]. It is recommended to be performed after 3 months of wearing complete dentures [4]. In contrast, the timing of functional assessment in implant denture wearers varies from several weeks to 10 years [5–16].

In a study [10] that investigated changes in masticatory function after wearing new dentures in patients with implant-supported dentures, it was reported that masticatory function improved or did not improve 20 days after wearing implants; however, it significantly improved 8 months after wearing. There is also a report that masticatory movement

did not significantly improve 6 months after wearing implant dentures [7]. These results suggest that it takes a considerable amount of time to adapt to changes in the oral environment and to acquire new masticatory functions after wearing implant dentures. In a study [16] that investigated the stability of the masticatory movement path and rhythm after implant treatment, the average value and standard deviation of each parameter were large 1 month after placement but gradually decreased from 6 months to 9 months after wearing implant dentures; almost the same values were maintained after 1 year. Based on these findings [1,7,10], a certain period is required to adjust to the masticatory function after wearing implant dentures, and a minimum of 8 months and preferably 1 year, is required to evaluate masticatory function after implant treatment.

One of the main goals of clinical dentistry is the restoration and maintenance of masticatory function. Therefore, to objectively evaluate the masticatory function, attempts have been made to examine masticatory performance [3–10,13–33], occlusal force [9,12,23,26,27,29,31,34–36], muscular activity [5,6,37–40], and masticatory movement [1,11–13,18,37,41–49]. Masticatory performance is the most widely used and various methods have been reported. Among these, the measurement of the glucose eluted from chewing gummy jelly is attracting attention because of its simple hygiene control and simple operation. It has been confirmed that there is a positive correlation between the masticatory performance measured by this method and the masticatory performance obtained by the sieving method [17].

Recently, evaluation of masticatory performance by measuring the amount of glucose eluted during the chewing of gummy jelly has been widely performed in healthy dentate adults [31,50,51], elderly adults [27,52], patients with metabolic disease [53], removable partial denture patients [24,28], removable complete denture patients [4,29], implant denture patients [23,54], patients with jaw deformity [30], mandibulectomy patients [33], and other groups.

The examination of masticatory movement is thought to be effective in quantitatively and objectively evaluating the masticatory function, and the amount of movement, rhythm, movement stability, and path patterns during mastication have been investigated. The movement of the mandibular incisal point during mastication is regular and stable in individual cycles in healthy adults, but irregular and unstable in patients with malocclusion or temporomandibular disorders (TMD) [42,43,45,55]. It has also been reported that the amount of movement (opening distance or masticatory width) increases, rhythm shortens, and path pattern changes from abnormal to normal after dental treatment [37,43,46,55]. These findings suggest that masticatory function can be assessed by analyzing masticatory movement.

There are two methods of recording mastication: free and unilateral. It has been reported that the movement itself changes greatly with the change from right to left and left to right during free chewing, so the movement is significantly more unstable than that during unilateral chewing [41]. It has also been reported that the masticatory performance is significantly higher on the unilateral chewing than on the free chewing [21]. It is recommended that unilateral chewing should be selected for evaluation of masticatory function [41].

Humans have laterality similar to the dominant hand, and there is a side, either left or right that is easier to use. The same is true for the oral cavity, with a side that is easy to use for chewing (habitual chewing side) and another that is not (non-habitual chewing side). Because there was a functional difference between the chewing sides in the stability of the movement [41], some studies have focused on the habitual chewing side for analysis [18,46]. Additionally, research on the evaluation of masticatory performance has been conducted on the habitual chewing side [4,19,22].

However, these are intended for dentate adults and it is unclear whether they can be applied to implant-supported denture wearers. It is also unknown whether patients with implants can recognize the habitual chewing side (the easy-to-chew side). Currently, most of the studies evaluating the masticatory function of implant-applied patients specify free chewing as the masticatory method or do not describe the masticatory method, and only

a few specify the habitual chewing side [20,25]. This is probably because the presence or absence of functional differences between the chewing sides in implant-applied patients has not been clarified.

Therefore, to clarify the functional differences between the chewing sides of implant-applied patients, the masticatory movement and masticatory performance of implant-supported denture wearers were compared between the right and left sides and between the habitual and non-habitual chewing sides.

## 2. Materials and Methods

This study was conducted in accordance with the principles of the Declaration of Helsinki. All experimental procedures were approved by the Institutional Review Board of the Faculty of Dentistry, The Nippon Dental University (Approval number: NDU-T2020-31). Informed consent was obtained from all participants after the purpose of the study was explained.

### 2.1. Participants

Forty-five adults (23 males, 22 females, 55–91 years old, mean 65.2 years old) participated in this study. Participants were adults with implant-supported dentures on both sides of their molars and had to meet the following inclusion and exclusion criteria. The sample size was calculated using a software program [56] (G*Power 3.1.9.3), with an $\alpha$ of 0.05, power of 0.8, and effect size of 0.5. A minimum of 34 participants were required to determine the presence or absence of differences between chewing sides in amount of movement, movement rhythm, and masticatory performance.

The inclusion criteria were (1) wearing implant dentures for at least 1 year, (2) no complaints about occlusion, (3) good general health or well-controlled systemic disease, and (4) an adequate cognitive function. The exclusion criteria were (1) clinical abnormalities of the masticatory system and (2) temporomandibular disorders (TMD) as defined by DC/TMD criteria [57].

### 2.2. Test Food

The test food (GLUCOLUMN, GC, Tokyo, Japan) was gummy jelly containing 5% glucose, with a cylindrical shape of 14 mm in diameter and 10 mm in height and weighing approximately 2 g. The other constituents of the gummy jelly were 41% reduced sugar syrup, 22% maltitol, 20% sorbitol, and 8% gelatin [32] (Figure 1).

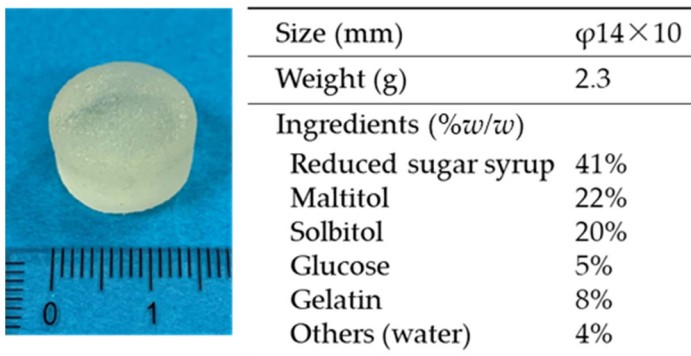

| Size (mm) | $\varphi 14 \times 10$ |
|---|---|
| Weight (g) | 2.3 |
| **Ingredients (%*w/w*)** | |
| Reduced sugar syrup | 41% |
| Maltitol | 22% |
| Solbitol | 20% |
| Glucose | 5% |
| Gelatin | 8% |
| Others (water) | 4% |

**Figure 1.** Test food (gummy jelly): Size, weight, and ingredients.

### 2.3. Experimental Design

Before the experiment, the participants were allowed to freely chew the test food, and the habitual chewing side was set by asking them to choose the side that was easier to chew [31].

The movement of the mandibular incisal point was recorded with a Motion Visitrainer (MVT V1, GC, Tokyo, Japan) when the participants chewed gummy jelly on one

side for 20 s. For 10 cycles from the fifth cycle after the start of mastication [58], the pattern of the movement path, the opening distance, the masticatory width, and the cycle time were calculated and used as parameters representing masticatory movement. The amount of glucose eluted when the participant chewed a gummy jelly for 20 s on one side was measured and was used as a parameter representing masticatory performance [32]. Each recording was performed twice, and the average was used for analysis [59].

### 2.4. Patterns of Masticatory Movement Path

For 10 cycles from the fifth to the fourteenth cycle after the start of mastication, the movement paths were superimposed, and the average path was displayed. The masticatory movement path patterns were classified into one of five types: N (Normal pattern) and A1 to A4 (Abnormal pattern) including four patterns consisting of a combination of two types for the opening path (o1: straight or concave, o2: convex) and two types for the closing path (c1: straight or convex, c2: concave) (N: o1–c1, A1: o1–c2, A2: o2–c1, A3: o2–c2), and a pattern (A4) in which the opening and closing crossed [48] (Figure 2).

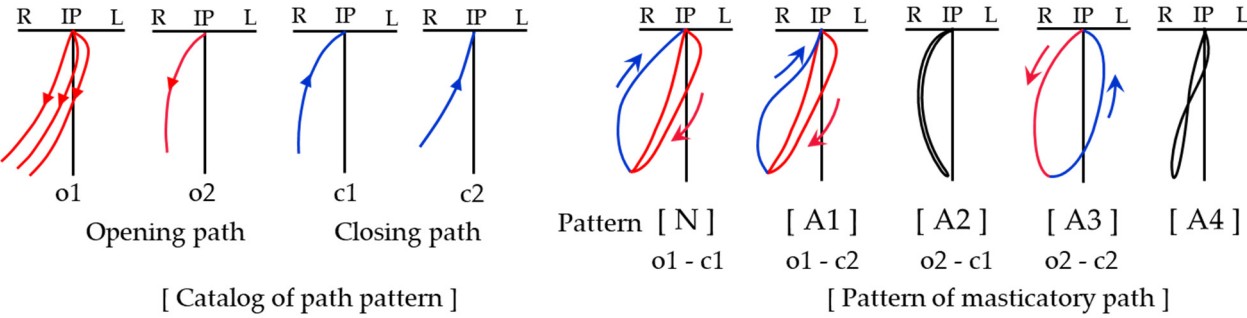

**Figure 2.** Catalog of path pattern and pattern of the masticatory path. IP: intercuspal position, R: right side, L: left side. N: normal pattern, A1–A4: abnormal pattern.

### 2.5. Calculation of Opening Distance, Masticatory Width, and Cycle Time

For 10 cycles from the fifth cycle, after calculating the average path from the opening and closing paths consisting of the vertical and lateral components of mandibular movement, the opening distance, and masticatory width were calculated. The opening distance was the vertical distance from the intercuspal position (IP) to the maximum mouth opening, and masticatory width was the average width of the first to ninth level (Figure 3 and Table 1). For 10 cycles from the fifth cycle, the opening time, closing time, occluding time, and sum of all three times were calculated as the cycle time. Then, the mean cycle time of the 10 cycles was calculated (Table 2).

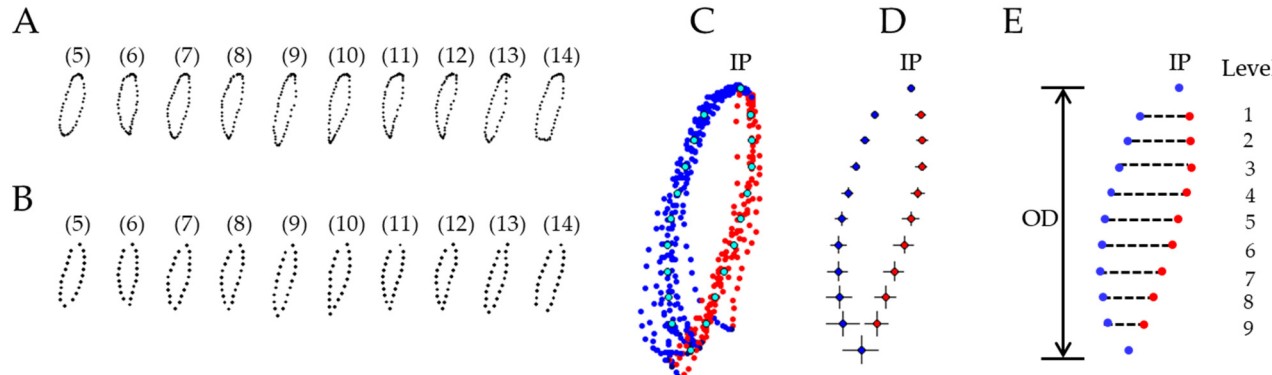

**Figure 3.** Method used to calculate average path, opening distance (OD), and masticatory width (example of participant 1). (**A**) 5–14 cycles on the right side chewing. (**B**) The coordinates for each cycle are determined by vertical division into 10 equally spaced sections. (**C**) Overlapping of each

cycle and average path. IP: intercuspal position. (**D**) Average path and standard deviations of each level. (**E**) OD and width from the first to the ninth level.

**Table 1.** Numerical data of the average path (example of participant 1).

| | | | | | | | | (mm) |
|---|---|---|---|---|---|---|---|---|
| **Level** | **Lateral Opening** | | **Lateral Closing** | | **Width** | | **Vertical** | |
| | **Mean** | **SD** | **Mean** | **SD** | | | **Mean** | **SD** |
| 0 | 0.0 | 0.1 | 0.0 | 0.1 | | | 0.0 | 0.0 |
| 1 | −0.7 | 0.3 | 2.3 | 0.2 | 3.0 | | 1.6 | 0.1 |
| 2 | −0.8 | 0.3 | 2.9 | 0.2 | 3.7 | | 3.3 | 0.1 |
| 3 | −0.7 | 0.3 | 3.5 | 0.3 | 4.2 | | 5.0 | 0.2 |
| 4 | −0.5 | 0.4 | 4.0 | 0.3 | 4.5 | | 6.7 | 0.3 |
| 5 | −0.1 | 0.5 | 4.4 | 0.4 | 4.5 | | 8.3 | 0.4 |
| 6 | −0.4 | 0.6 | 4.6 | 0.5 | 4.2 | | 10.0 | 0.5 |
| 7 | 1.0 | 0.6 | 4.6 | 0.6 | 3.6 | | 11.7 | 0.6 |
| 8 | 1.6 | 0.6 | 4.6 | 0.8 | 3.0 | | 13.4 | 0.7 |
| 9 | 2.1 | 0.8 | 4.3 | 1.0 | 2.2 | | 15.0 | 0.8 |
| 10 | 3.2 | 1.1 | 3.2 | 1.1 | | | 16.7 [a] | 0.8 |
| Mean | | 0.56 | | 0.55 | 3.6 [b] | | | |

[a]: Opening distance (OD), [b]: Masticatory width.

**Table 2.** Numerical data of the opening, closing, occluding, and cycle times (example of participant 1).

| | | | | (ms) |
|---|---|---|---|---|
| **Cycle** | **Opening Time** | **Closing Time** | **Occluding Time** | **Cycle Time** |
| 1 | 200 | 220 | 160 | 580 |
| 2 | 190 | 230 | 170 | 590 |
| 3 | 180 | 250 | 160 | 590 |
| 4 | 170 | 220 | 170 | 560 |
| 5 | 190 | 230 | 150 | 570 |
| 6 | 220 | 210 | 160 | 590 |
| 7 | 170 | 240 | 140 | 550 |
| 8 | 180 | 220 | 170 | 570 |
| 9 | 180 | 220 | 160 | 560 |
| 10 | 170 | 240 | 170 | 580 |
| Mean | 185 | 228 | 161 | 573 |

*2.6. Measurement of Masticatory Performance*

Participants were asked to chew the test food on one side for 20 s without swallowing. After chewing, they were asked to hold 10 mL of water in their mouth for a moment and to spit into a cup with a filter. The glucose concentration of the filtrate was measured using a glucose-measuring device (GS-2, GC, Tokyo, Japan). The measured value was taken as a parameter of masticatory performance (Figure 4) [32]. It has been confirmed that the effect of saliva on the measurement results is less than 5% when holding 10 mL of water and spitting with the gummy jelly [60].

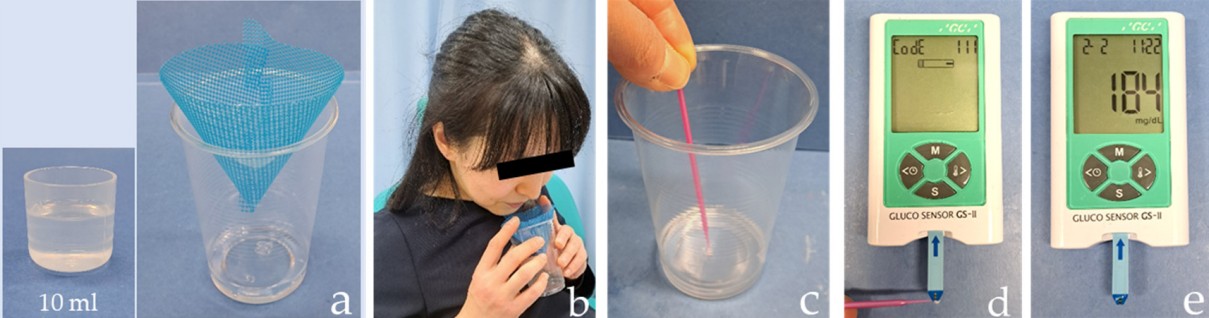

**Figure 4.** Procedures for measuring masticatory performance. (**a**) Water preparation: Prepare 10 mL of water in a cup and sieve. Have the participant chew the gummy jelly on the habitual chewing side for 20 s. (**b**) After chewing, have the participant hold 10 mL of water in the mouth and then spit it out along with the gummy jelly. (**c**–**e**) Collect the filtrate with a brush, and place it on the sensor tip. The glucose concentration will be displayed a few seconds later using a glucose-measuring device (GS-2, GC, Tokyo, Japan). This measured value is the amount of glucose eluted.

### 2.7. Statistical Analysis

All data were analyzed using statistical software (SPSS version 27.0, IBM Corp., Armonk, NY, USA). After classifying the movement path patterns of all participants into five types (N, A1 to A4), the frequency of normal (N) and abnormal (A1 to A4) patterns in the right and left chewing sides and the habitual and non-habitual chewing sides was investigated using the chi-squared test. Next, the parameter values representing masticatory movement (opening distance, masticatory width, and cycle time) and masticatory performance (amount of glucose eluted) were compared between the right and left chewing sides and between the habitual and non-habitual chewing sides. Comparisons between chewing sides were performed using a paired t-test after confirming the normality of the data using the Shapiro–Wilk test. All statistical analyses were performed with the significance level set at $p$ values of 0.05.

## 3. Results

### 3.1. Masticatory Path Patterns

Figure 5 shows examples of masticatory movement path for some participants. Table 3 shows the masticatory path pattern for all participants. Table 4 shows the frequency of masticatory path patterns. When observing the right and left chewing sides, there was no difference in the frequency of masticatory path patterns between the chewing sides ($x^2 = 0.039$, $p = 1.000$). Most participants had a normal pattern on the habitual chewing side (normal: 93%, abnormal: 7%); however, abnormal patterns were observed on the non-habitual chewing side (normal: 47%, abnormal: 53%). There was a significant difference in the pattern frequency between the chewing sides ($x^2 = 23.333$, $p < 0.001$).

**Table 3.** Masticatory path pattern for all participants.

| Participant | 1–12 | 13–34 | 35–42 | 43–45 |
|---|---|---|---|---|
| Habitual | N | N | N | A |
| Non-habitual | N | A | A | A |
| Right side | N | N | A | A |
| Left side | N | A | N | A |

N: Normal pattern, A: Abnormal pattern.

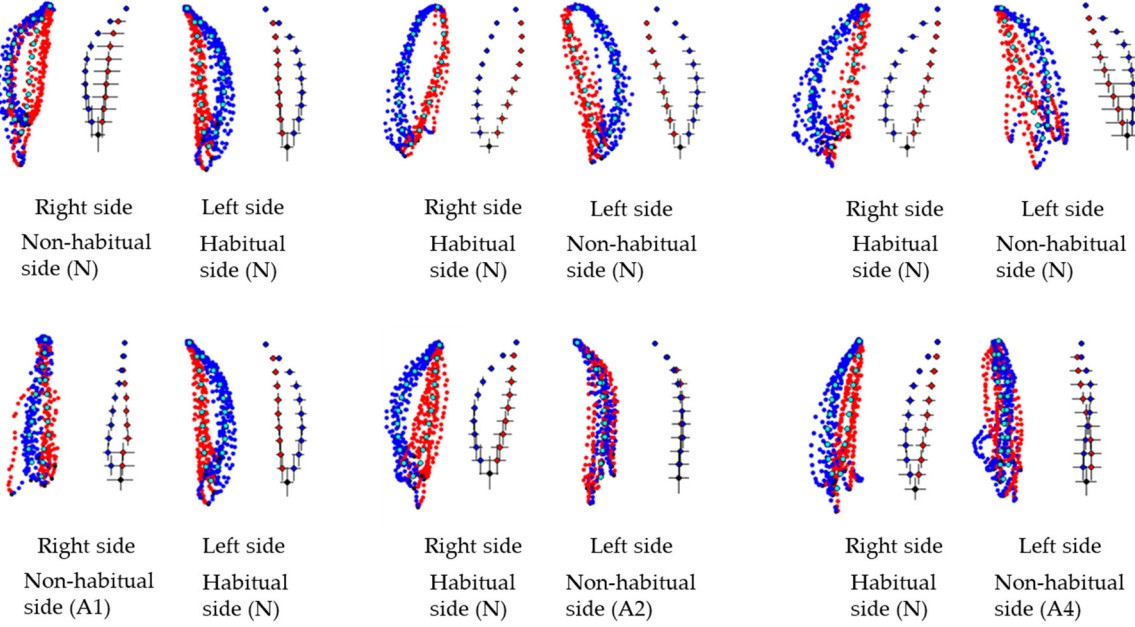

**Figure 5.** Examples of masticatory movements path for some participants.

**Table 4.** Frequency of masticatory path patterns.

|          | Right Side | | Left Side | | Habitual Side | | Non-Habitual Side | |
|----------|------|------|------|------|------|------|------|------|
|          | n | % | n | % | n | % | n | % |
| Normal   | 33 | 73.3 | 30 | 66.7 | 42 | 93.3 | 21 | 46.7 |
| Abnormal | 12 | 26.7 | 15 | 33.3 | 3 | 6.7 | 24 | 53.3 |

*3.2. Masticatory Movement*

The Shapiro–Wilk test confirmed normality of the data (opening distance, $p = 0.306$–$0.606$; masticatory width, $p = 0.230$–$0.290$; cycle time, $p = 0.158$–$0.476$).

Table 5 shows the means and standard deviations for the values of parameters representing masticatory movement. When comparing the right and left chewing sides, there was no significant difference between the chewing sides in terms of opening distance, masticatory width, cycle time (opening distance, $p = 0.471$; masticatory width, $p = 0.350$; cycle time, $p = 0.104$). When comparing the habitual and non-habitual chewing sides, the habitual chewing side showed a larger opening distance and masticatory width, a shorter cycle time, and significant differences were observed between the chewing sides (opening distance, $p < 0.001$; masticatory width, $p = 0.008$; cycle time, $p = 0.004$).

**Table 5.** Means and standard deviations for the values of the movement path and movement rhythm.

|                      | Right Side | Left Side | *p*-Value |
|----------------------|------------|-----------|-----------|
| Movement path (mm)   |            |           |           |
| Opening distance     | $15.0 \pm 1.7$ | $15.4 \pm 2.0$ | 0.471 |
| Masticatory width    | $2.4 \pm 0.7$ | $2.2 \pm 0.7$ | 0.350 |
| Movement rhythm (ms) |            |           |           |
| Cycle time           | $693.3 \pm 82.7$ | $686.2 \pm 69.4$ | 0.104 |
|                      | **Habitual Side** | **Non-Habitual Side** | ***p*-Value** |
| Movement path (mm)   |            |           |           |
| Opening distance     | $15.9 \pm 1.7$ | $14.4 \pm 1.6$ | <0.001 |
| Masticatory width    | $2.5 \pm 0.7$ | $2.1 \pm 0.7$ | 0.008 |
| Movement rhythm (ms) |            |           |           |
| Cycle time           | $675.3 \pm 71.8$ | $703.3 \pm 82.3$ | 0.004 |

*3.3. Masticatory Performance*

The Shapiro–Wilk test confirmed normality of the data ($p$ = 0.243–0.575).

Table 6 shows the means and standard deviations for the values of parameters representing masticatory performance. When comparing the right and left chewing sides, the amount of glucose eluted was similar, and no significant difference was observed between chewing sides ($p$ = 0.951). When comparing the habitual and non-habitual chewing sides, the amount of glucose eluted was larger on the habitual chewing side than on the non-habitual chewing side, and a significant difference was observed between the chewing sides ($p$ < 0.001).

**Table 6.** Means and standard deviations for the values of the masticatory performance.

| | **Right Side** | **Left Side** | *p*-**Value** |
|---|---|---|---|
| Masticatory performance (mg/dL) | 150.7 ± 31.9 | 151.1 ± 29.7 | 0.951 |
| | **Habitual Side** | **Non-Habitual Side** | *p*-**Value** |
| Masticatory performance (mg/dL) | 162.6 ± 32.4 | 139.2 ± 23.6 | <0.001 |

**4. Discussion**

*4.1. Determination of Habitual Chewing Side*

To determine the habitual chewing side, a method based only on interviews [4,17,34,61], based on interviews after actual mastication [1,27,48], measuring from the beginning of the masticatory cycle [44], using EMG recording of the masseter muscle during mastication [38–40,62], and methods using jaw movement patterns [47] have been re-ported. Among them, the simplest method, which is to decide by interview, is based only on the subjectivity of the participants, and it is easily influenced by the dominant hand [34,39]. The authors found that when participants were asked about their habitual chewing side only in the interview, they often answered the dominant side, and the answers changed when the question was asked after free chewing. Therefore, we used a method in which the test food was chewed freely before the experiment, the side that was easy to chew was confirmed, and the habitual chewing side was set [26,32,36]. Therefore, this method was adopted in this study. It has also been reported that by allowing the test food to be chewed freely before the experiment, it becomes accustomed to chewing the test food, and the inter-measurement variation is reduced; therefore, even a single measurement is sufficient [31]. However, we believed that multiple measurements would be better, and in this study, the average of two measurements was used as the parameter value.

*4.2. Movement Path Pattern between Chewing Sides*

Regarding the movement path pattern, it has been reported that there are many nor-mal patterns in healthy adults [48], and many abnormal patterns in adults with mal-occlusion; however, they recover to normal patterns after treatment [43], and abnormal pat-terns appear when experimental occlusal interference is applied to healthy adults [49]. In this study, most of the participants had a normal pattern on the habitual chewing side; however, a relatively more abnormal pattern was observed on the non-habitual chewing side. Based on research reports on malocclusion and experimental occlusal interference, it is highly likely that some occlusal abnormality exists on the non-habitual chewing side. Therefore, it can be said that there is a need for inspection of occlusion on the non-habitual chewing side and further occlusal adjustment.

*4.3. Masticatory Function between Chewing Sides*

It has been reported that masticatory function deteriorates due to tooth loss and that dental prosthetic treatment increases the amount of masticatory movement, shortens the cycle time, and improves masticatory performance [4,28,46]. Similar findings have been reported for implant denture treatment [6,7,11,14–16,23]. These results indicate that the

amount of movement, rhythm, and masticatory performance can be used to evaluate masticatory function.

A study investigating the functional differences between the chewing sides of dentate adults reported that the masticatory movement path on the habitual chewing side was significantly stable [41] and the masticatory performance was higher than that on the non-habitual chewing side [32]. Therefore, the number of studies targeting the habitual chewing side has recently increased [29,63,64]. However, no studies have investigated functional differences between the chewing sides of implant denture wearers, and only a few studies have analyzed the habitual chewing side [20,25]. As mentioned above, only a few studies have described masticatory methods [8,65]. In this study, the amount of movement was greater, the movement rhythm was shortened, and masticatory performance was higher on the habitual chewing side than that on the non-habitual chewing side, indicating a significant difference between the chewing sides. These results show that even implant denture wearers recognize differences in chewing ease between chewing sides and that implant-supported denture wearers have a larger and faster masticatory movement and higher masticatory performance on the easy-to-chew side (habitual chewing side) than on the difficult-to-chew side (non-habitual chewing side). Additionally, since a significant difference was observed between the chewing sides, the habitual chewing side should be analyzed in implant-supported denture wearers as well.

### 4.4. Limitations

The participants in this study had similar occlusal conditions on the right and left sides. Therefore, it included those with implant-fixed dentures in both the upper and lower jaws, those with complete dentures in the upper jaw and implant denture in the lower jaw, and those with natural dentition in the upper jaw and implant dentures in both molars of the lower jaw. Since the participants with similar occlusal conditions on the right and left sides were selected, it was considered that there was no problem in comparing the habitual chewing side and the non-habitual chewing side. However, when comparing by individual implants denture, it may be necessary to increase the number of participants and investigate participants with the same maxillary and mandibular conditions. Furthermore, further study may be necessary after taking into account the effects of conditions such as the condition of the oral cavity [66] and the condition of the implant [67].

### 5. Conclusions

To clarify the functional differences between the chewing sides of implant-supported denture wearers, the masticatory movement and masticatory performance of 45 patients who had implants applied to their bilateral molars were compared between the chewing sides. Consequently, the patients with implants had greater and faster masticatory movements and higher masticatory performance on the habitual chewing side than that on the non-habitual chewing side. These results suggest that there is a functional difference between habitual and non-habitual chewing sides in masticatory movement and the masticatory performance of implant-supported denture wearers.

**Author Contributions:** Conceptualization, M.Y., H.S. and S.O.; methodology, M.S., M.K. and H.U.; formal analysis, M.S., M.K., H.U., K.H. and Y.M.; investigation, S.O., M.S., H.T., K.H. and Y.M.; data curation, M.Y., H.U., M.S. and M.K.; writing—original draft preparation, M.Y. and H.S.; writing—review and editing, M.Y., H.S., S.O. and M.K.; visualization, M.S., M.K., H.U., K.H. and Y.M.; supervision, M.Y., H.S. and H.T.; project administration, H.S.; All authors have read and agreed to the published version of the manuscript.

**Funding:** This research received no external funding.

**Institutional Review Board Statement:** The study was conducted according to the guidelines of the Declaration of Helsinki and approved by Institutional Review Board of the Faculty of Dentistry, The Nippon Dental University (approval number: NDU–T2012–29).

**Informed Consent Statement:** Informed consent was obtained from all subjects involved in the study.

**Data Availability Statement:** Not applicable.

**Conflicts of Interest:** The authors declare no conflict of interest.

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
