# Peer review of "Functional Differences between Chewing Sides of Implant-Supported Denture Wearers"

_prosthesis, doi:10.3390/prosthesis5020025_

Round 1

Reviewer 1 Report

The examined group of patients was not homogenous in terms of dentition, because it is totally different have dentated maxilla or to have complete denture from the functional point of view. So it would be interesting to have the research design on a homogenous group of patients for such a study.

Authors should to add information about picture presented the gummy jelly (fig. 3 a) in the line 108 to 111.

To have the methods clear and repeatable I would suggest writing, that the patients were ask not to swallow during the test with chew the jelly.

Was the saliva stimulation measured? Has it the influence to the results of the amount of glucose received after chewing? The amount of glucose could be related with the saliva stimulation without any chewing.

Point 2.6 – it should be written in the text not only as the caption under the picture, that the participants were ask to “wash” the oral cavity with 10 ml of water after the chewing test.

 Line 229 – capital letter is by mistake

Discussion – line 239 to 260 I suggest to transfer to the introduction, because it was not the subject of examination, but the important for study design.

Author Response

Reviewer 1

Comments and Suggestions for Authors

The examined group of patients was not homogenous in terms of dentition, because it is totally different have dentated maxilla or to have complete denture from the functional point of view. So it would be interesting to have the research design on a homogenous group of patients for such a study.

Response

Thank you for a sincere review of the article that we submitted.

Your comment has been helpful in allowing us to revise our manuscript. 

We have revised the paper in accordance with the comments made by you.

Comment 1

Authors should to add information about picture presented the gummy jelly (fig. 3 a) in the line 108 to 111.

Response

Thank you for your comment.

We added the information about the gummy jelly.

Comment 2

To have the methods clear and repeatable I would suggest writing, that the patients were ask not to swallow during the test with chew the jelly.

Was the saliva stimulation measured? Has it the influence to the results of the amount of glucose received after chewing? The amount of glucose could be related with the saliva stimulation without any chewing.

Point 2.6 – it should be written in the text not only as the caption under the picture, that the participants were ask to “wash” the oral cavity with 10 ml of water after the chewing test.

Response

Thank you for your comment.

We added the following sentences according to the comment.

2.6. Measurement of masticatory performance

Participants were asked to chew the test food on one side for 20 seconds without swallowing. After chewing, they were asked to hold 10 ml of water in their mouth for a moment and to spit into a cup with a filter. The glucose concentration of the filtrate was measured using a glucose-measuring device (GS-2, GC, Tokyo, Japan). The measured value was taken as a parameter of masticatory performance (Figure 3) [32]. It has been confirmed that the effect of saliva on the measurement results is less than 5% by holding 10 ml of water and spit with the gummy jelly [59].

Comment 3

 Line 229 – capital letter is by mistake

Response

Thank you for your comment.

We modified according to the comment.

Comment 4

Discussion – line 239 to 260 I suggest to transfer to the introduction, because it was not the subject of examination, but the important for study design.

Response

Thank you for your comment.

We transferred according to the comment.

Reviewer 2 Report

The present study aims to clarify the presence or absence of functional differences between the chewing sides in implant-supported denture wearers. The results highlighted a functional difference between the habitual and non-habitual chewing sides in the masticatory movement and masticatory performance of implant-supported denture wearers. The manuscript is well structured, with the main purpose so clear. 

However, I have several suggestions. 

Introduction:
Line 54-56: The movement of the mandibular incisal point during mastication is regular and stable in individual cycles in healthy adults and irregular and unstable in patients with malocclusion patients or temporomandibular disorders (TMD) patients – please correct this sentence. 

Material and Methods:
- Line 100-101: Exclusion criteria: – 1) clinical abnormalities of the masticatory system and 2) signs and 100 symptoms of TMD and maxillofacial pain - please indicate on what basis were patients tested for TMD? Was the RDC/TMD protocol used?

Discussion: 

Line: 261-265: To determine the habitual chewing side, a method based only on interviews, based 261 on interviews after actual mastication, measuring from the beginning of the masticatory cycle, using EMG recording of the masseter muscle during mastication, and methods using jaw movement patterns have been reported – please provide citations where appropriate

Overall:
In general, the work is interesting and can contribute to the literature. I hope my suggestions will help improve this work.

Author Response

Reviewer 2

The present study aims to clarify the presence or absence of functional differences between the chewing sides in implant-supported denture wearers. The results highlighted a functional difference between the habitual and non-habitual chewing sides in the masticatory movement and masticatory performance of implant-supported denture wearers. The manuscript is well structured, with the main purpose so clear. 

However, I have several suggestions.

Response

Thank you for a sincere review of the article that we submitted.

Your comment has been helpful in allowing us to revise our manuscript. 

We have revised the paper in accordance with the comments made by you.

Comment 1

Introduction:

Line 54-56: The movement of the mandibular incisal point during mastication is regular and stable in individual cycles in healthy adults and irregular and unstable in patients with malocclusion patients or temporomandibular disorders (TMD) patients – please correct this sentence.

Response

Thank you for your comment.

We modified as below according to the comment.

The movement of the mandibular incisal point during mastication is regular and stable in individual cycles in healthy adults, but irregular and unstable in patients with malocclusion patients or temporomandibular disorders (TMD).

Comment 2

Material and Methods:

Line 100-101: Exclusion criteria: – 1) clinical abnormalities of the masticatory system and 2) signs and symptoms of TMD and maxillofacial pain - please indicate on what basis were patients tested for TMD? Was the RDC/TMD protocol used?

Response

Thank you for your comment.

We modified as below according to the comment.

The exclusion criteria were: 1) clinical abnormalities of the masticatory system and 2) temporomandibular disorders (TMD) as defined by RDC-TMD criteria [57].

Comment 3

Discussion: 

Line: 261-265: To determine the habitual chewing side, a method based only on interviews, based on interviews after actual mastication, measuring from the beginning of the masticatory cycle, using EMG recording of the masseter muscle during mastication, and methods using jaw movement patterns have been reported – please provide citations where appropriate

Response

Thank you for your comment.

We added the appropriate citation as below according to the comment.

To determine the habitual chewing side, a method based only on interviews [3,17,34,60], based on interviews after actual mastication [16,27,48], measuring from the beginning of the masticatory cycle [44], using EMG recording of the masseter muscle during mastication [38-40,61], and methods using jaw movement patterns [47] have been reported. Among them, the simplest method, which is to decide by interview, is based only on the subjectivity of the participants, and it is easily influenced by the dominant hand [34,39].

Reviewer 3 Report

Methodological Biases exist

(The Authors must see my remarks)

Author Response

Reviewer 3

Response

Thank you for a sincere review of the article that we submitted.

Your comment has been helpful in allowing us to revise our manuscript. 

We have revised the paper in accordance with the comments made by you.

Comment 1

Please clarify the type of the article, eg. Research?

Response

Thank you for your comment.

Since this paper is an original research manuscript, we think that 'Article' is appropriate.

Comment 2

Line: 15 Replace by ''45''....

Response

Thank you for your comment.

We think that the beginning of the sentence is correct “Forty-five”, not “45”.

Comment 3

Line: 24 ''p-value''?

Line: 27 ''p-value''?

Response

Thank you for your comment.

We added ''p-value'' according to the comment.

Comment 4

Line: 97 How did the Authors determine the study size? Protocol? Preference(s)?

Response

Thank you for your comment.

The study size was determined (34 persons or more) by calculating with the software[56](G*Power 3.1.9.3).

We modified as below according to the comment.

2.1. Participants

Forty-five adults (23 males, 22 females, 55–91 years old, mean 65.2 years old) participated in this study. Participants were adults with implant-supported dentures on both sides of their molars and had to meet the following inclusion and exclusion criteria. The sample size was calculated using a software program[56] (G*Power 3.1.9.3), with an α of 0.05, power of 0.8, and effect size of 0.5, requiring 34 participants.

Comment 5

Line: 98 References?

Line: 101 References?

Line: 103 References?

Response

Thank you for your comment.

These are the inclusion and exclusion criteria for this study, and the minimum number of participants required, and we do not think the reference is necessary.

Comment 6

Line 108 Do not state trade-marks....

Line: 118 Do not state trade-marks....

Line 174: Do not state trade-marks....

Response

Thank you for your comment.

Trademarks are usually asked to be described. Other articles in your journal also described the trademark.

Comment 7

Line: 110 References?

Line 142: References?

Response

Thank you for your comment.

We added the citation according to the comment.

Comment 8

Line 189: Weak and unreliable model... The Authors should use a Logistic Regression analysis model....

Line192: Was the distribution a normal one? Moreover, the study size is extremely low.....

Response

Thank you for your comment.

This study covers 45 people with implant dentures on both molars, so there is no problem with the sample size.

We think that because Table 1 shows the composition of the participants, it was misunderstood that the sample size was small.

We will delete Table 1 so as not to mislead the reader.

We were also doing a normality test, so we added that as below to the results.

3.2. Masticatory movement

The Shapiro–Wilk test confirmed normality of the data (opening distance, P = 0.306–0.606; masticatory width, P = 0.230–0.290; cycle time, P = 0.158–0.476).

3.3. Masticatory performance

The Shapiro–Wilk test confirmed normality of the data (P=0.243-0.575).

Comment 9

Line 254: By whom?

Response

Thank you for your comment.

We modified as below according to the comment.

In a study [16] that investigated the stability of the masticatory movement path and rhythm after implant treatment, the average value and standard deviation of each parameter were large 1 month after placement but gradually decreased from 6 months to 9 months after wearing implant dentures; almost the same values were maintained after 1 year.

Comment 10

Line 264: References?

Response

Thank you for your comment.

We added the appropriate citation as below according to the comment.

To determine the habitual chewing side, a method based only on interviews [3,17,34,60], based on interviews after actual mastication [16,27,48], measuring from the beginning of the masticatory cycle [44], using EMG recording of the masseter muscle during mastication [38-40,61], and methods using jaw movement patterns [47] have been reported. Among them, the simplest method, which is to decide by interview, is based only on the subjectivity of the participants, and it is easily influenced by the dominant hand [34,39].

Comment 11

Line 299: Do not repeat data....

Response

Thank you for your comment.

We consider these to be necessary sentences, so please leave them as they are.

Comment 12

Line 309: Limitations?

Response

Thank you for your comment.

We added the following sentences to the end of discussion.

The participants in this study had similar occlusal conditions on the right and left sides. Therefore, it included those with implant-fixed denture in both the upper and lower jaws, those with complete dentures in the upper jaw and implant dentures in the lower jaw, and those with natural dentition in the upper jaw and implant dentures in both molars of the lower jaw. Since the participants with similar occlusal conditions on the right and left sides were selected, it was considered that there was no problem in comparing the habitual chewing side and the non-habitual chewing side. However, it may be necessary to increase the number of participants and investigate participants with the same maxillary and mandibular conditions.

Reviewer 4 Report

Dears. This article is correctly done and very interesting for the authors. Comparing both sides after implant rehabilitation gives us important information. 

Reading the article, I find bibliographical references missing in some parts of the introduction and discussion. For example, in lines 70-71, 77-79, 254-256, 261-264. There are many sentences without the corresponding reference.

A proposal. Using the same methodology it would be interesting to take patients with missing teeth on the usual chewing side and rehabilitate them with implants and check if they recover the function prior to tooth loss.

Thanks

Author Response

Reviewer 4

Dears. This article is correctly done and very interesting for the authors. Comparing both sides after implant rehabilitation gives us important information.

Response

Thank you for a sincere review of the article that we submitted.

Your comment has been helpful in allowing us to revise our manuscript. 

We have revised the paper in accordance with the comments made by you.

Comment 1

Reading the article, I find bibliographical references missing in some parts of the introduction and discussion. For example, in lines 70-71, 77-79, 254-256, 261-264. There are many sentences without the corresponding reference.

Response

We added the appropriate citation according to the comment.